# Phosphoinositide 3-Kinase P110δ-Signaling Is Critical for Microbiota-Activated IL-10 Production by B Cells that Regulate Intestinal Inflammation

**DOI:** 10.3390/cells8101121

**Published:** 2019-09-21

**Authors:** Akihiko Oka, Yoshiyuki Mishima, Bo Liu, Jeremy W. Herzog, Erin C. Steinbach, Taku Kobayashi, Scott E. Plevy, R. Balfour Sartor

**Affiliations:** 1Center for Gastrointestinal Biology and Disease, Department of Medicine, Division of Gastroenterology and Hepatology, University of North Carolina at Chapel Hill, Chapel Hill, NC 27599, USA; akihiko_oka@med.unc.edu (A.O.); mtmtyui@med.shimane-u.ac.jp (Y.M.); bo_liu@med.unc.edu (B.L.); jeremy_herzog@med.unc.edu (J.W.H.); erin_klein@med.unc.edu (E.C.S.); scott.plevy@synlogictx.com (S.E.P.); 2Department of Internal Medicine II, Shimane University Faculty of Medicine, Izumo, Shimane 693-8501, Japan; 3Division of Allergy, Immunology and Rheumatology, Department of Pediatrics, University of North Carolina School of Medicine, Chapel Hill, NC 27599, USA; 4Department of Microbiology and Immunology, University of North Carolina at Chapel Hill, Chapel Hill, NC 27599, USA; kobataku@insti.kitasato-u.ac.jp; 5Center for Advanced IBD Research and Treatment, Kitasato University Kitasato Institute Hospital, Minato-ku, Tokyo 108-8642, Japan; 6Synlogic Therapeutics, Boston, MA 02139, USA

**Keywords:** PI3Kδ, inflammatory bowel diseases, regulatory B cells, IL-10, residential microbiota

## Abstract

The phosphoinositide 3-kinase catalytic subunit p110δ (PI3Kδ) gene maps to a human inflammatory bowel diseases (IBD) susceptibility locus, and genetic deletion of PI3Kδ signaling causes spontaneous colitis in mice. However, little is known regarding the role of PI3Kδ on IL-10-producing B cells that help regulate mucosal inflammation in IBD. We investigated the role of PI3Kδ signaling in B cell production of IL-10, following stimulation by resident bacteria and B cell regulatory function against colitis. In vitro, B cells from PI3Kδ^D910A/D910A^ mice or wild-type B cells treated with PI3K specific inhibitors secreted significantly less IL-10 with greater IL-12p40 following bacterial stimulation. These B cells failed to suppress inflammatory cytokines by co-cultured microbiota-activated macrophages or CD4^+^ T cells. In vivo, co-transferred wild-type B cells ameliorated T cell-mediated colitis, while PI3Kδ^D910A/D910A^ B cells did not confer protection from mucosal inflammation. These results indicate that PI3Kδ-signaling mediates regulatory B cell immune differentiation when stimulated with resident microbiota or their components, and is critical for induction and regulatory function of IL-10-producing B cells in intestinal homeostasis and inflammation.

## 1. Introduction

Inflammatory bowel diseases (IBD), represented by Crohn’s disease and ulcerative colitis, are chronic intestinal immune disorders characterized by dysregulated immune response to enteric resident bacteria in genetically susceptible hosts [1,2,3]. Resident bacteria induce pro-inflammatory cytokines such as IL-12p40 and TNF-α from host cells, which are the main therapeutic targets of most current IBD therapies [1,3,4]. However, the same microorganisms can also induce anti-inflammatory responses in normal hosts [1,2,3,5,6]. A delicate, homeostatic balance between anti-inflammatory and pro-inflammatory cytokines is maintained in healthy individuals, while dysregulated immune balances against resident bacteria are a key factor in the pathogenesis of IBD [1,2,3,5,7].

The anti-inflammatory cytokine IL-10 is secreted by many types of intestinal cells, including T cells, B cells, macrophages, dendritic cells and epithelial cells [4,8,9,10,11]. Previous studies have mainly focused on T cells and macrophages to determine the roles of IL-10 in the pathogenesis of IBD [1,3,4,12], while our work and that of others has demonstrated the importance of IL-10 production by B cells in experimental colitis [10,11,13,14,15,16,17]. Depletion of B cells contributes to the development of human IBD [15,16] and enhanced inflammation in a murine model of chronic colitis [14]. We have also shown that IL-10-producing B cells induced by enteric bacteria maintain mucosal homeostasis and protect against colonic inflammation [6,11]. Those observations indicate that intestinal B cells play a protective role through IL-10 production to maintain mucosal homeostasis. However, the detailed mechanisms underlying induction, function and maintenance of bacteria-activated IL-10-producing B cells are not fully understood.

Microarray analysis comparing IL-10-producing with non-producing splenic B cells has revealed that phosphoinositide-3 kinases (PI3Ks)-Akt signaling is a key pathway of IL-10 production [18]. Activation of PI3Ks signaling in B cells suppressed allergic diseases through IL-10 production [18]. PI3Ks are recognized as a critical regulator of cell growth and immune function [19]. Class IA PI3Ks, which are heterodimeric enzymes consisting of a catalytic subunit (p110α, p110β or p110δ) and a regulatory subunit (p85, p55 or p50), regulate toll-like receptor (TLR) signaling in both positive and negative ways [12,20,21,22,23,24,25] in Myd88-dependent and independent manners [26]. The PI3K subunit p110δ (PI3Kδ) is highly expressed in leukocytes, whereas p110α and p110β are expressed ubiquitously [27]. Colonic PI3Kδ is induced immediately after TLR activation by enteric bacteria [12,23,24]. Together, these data suggest PI3Kδ is a gate-keeper to control immune responses to enteric bacteria and is an early event in TLR signaling [23].

As for PI3Kδ in gastrointestinal diseases, the human PI3Kδ gene (encoded by *PIK3CD*) maps to the IBD7 susceptibility locus [28,29] and a loss of PI3Kδ leads to defective B cell function and intestinal inflammation in humans [30]. Similarly, a loss-of-function mutation of PI3Kδ gene induces spontaneous colitis through Th1/Th17 activation in mice [12,31,32]. Although the mechanisms by which PI3Kδ dysfunction induces colitis remain unclear, reduced IL-10 production by bacteria-stimulated macrophages and regulatory T cells potentially contribute to induce colitis in PI3Kδ mutant mice [12,32,33]. Conversely, B cells are a major source of resident bacteria-stimulated IL-10 production in the colon, and IL-10-producing B cells play a pivotal role in maintaining mucosal homeostasis [6,11], while the involvement of PI3Kδ-signaling in B cell IL-10-production is not fully understood. Thus, we hypothesized that PI3Kδ-signaling impacts bacteria-stimulated IL-10 producing B cells and regulates intestinal inflammation. Macrophages and effector CD4^+^ T cells play a critical role in intestinal inflammation [1,3,4]. To explore this hypothesis, we examined cytokine profiles of in vitro co-cultured macrophages or CD4^+^ T cells with PI3Kδ-mutant B cells. In addition, we investigated anti-inflammatory effects of PI3Kp110δ on B cells by in vivo adoptive transfer of PI3Kp110δ-mutant B cell into a T cell-mediated colitis model.

## 2. Materials and Methods

### 2.1. Mice

C57BL/6J background wild-type (WT), IL-10-deficient (*Il10*^−/−^) and Rag2-deficient (*Rag2*^−/−^) mice were obtained from Jackson laboratory (Bar Harbor, ME, USA). We generated *Il10*^−/−^; *Rag2*^−/−^ double-deficient C57BL/6J mice by crossing *Il10*^−/−^ with *Rag2*^−/−^ mice. PI3Kp110δ^D910A/D910A^ mutant (PI3Kδ^D910A^) mice on the C57BL/6J background, which have a loss-of-function (catalytically inactive) point mutation in a gene encoding the PI3K catalytic subunit p110δ, were previously obtained from Dr. B. Vanhaesebroeck (Queen Mary University of London, London, UK) [31]. *Il10*-eGFP-reporter (WT; *Il10*^eGFP^) mice on the C57BL/6J background, which exhibit normal IL-10 secretion ability, were provided by Dr. C. L. Karp (Global Health, Bill and Melinda Gates Foundation, Seattle, WA, USA) [34]. *Il10*-eGFP-reporter; PI3Kδ^D910A^ mice (PI3Kδ^D910A^; *Il10*^eGFP^) on a C57BL/6J background were generated by crossing PI3Kδ^D910A^ with WT; *Il10*^eGFP^ mice. All mice were bred and housed in specific-pathogen-free (SPF) conditions at the University of North Carolina at Chapel Hill (UNC). Age and numbers of used mice are indicated in figures or legends. All animal procedures were approved by the UNC Institutional Animal Care and Use Committee. Genotyping of deficient/mutant/transgenic mice were performed by PCR using the following oligonucleotides: *Il10* 5’, 5′-ACCAAGGTGTCTACAAGGCCATGAATGAATT-3’; GFP 5’, 5’-GAGGAAATTGCATCGCATTGTCTGAGTAGGT-3’; *Il10* 3’, 5’-CAAAGGCAGACAAACAATACACCATTCCCA-3’, PI3Kδ common, 5’- CCTGCACAGAAATGCACTTCC-3’; PI3Kδ wild-type reverse, 5’-AACGAAGCTCTCAGAGAAAGCTGG-3’; PI3Kδ^D910A^ mutant reverse, 5’-CGCCTTCTATCGCCTTCTTGAC-3’. *Rag2* common, 5’-CAGCGCTCCTCCTGATACTC-3’; *Rag2* wild-type reverse, 5’-TGCATTCCTAGAGCGTCCTT-3’; *Rag2* mutant reverse, 5’-GGTCATCCTTTGCAACACAG-3’, *Il10* common, 5’-CTTGCACTACCAAAGCCACA-3’; *Il10* wild-type reverse, 5’-GTTATTGTCTTCCCGGCTGT-3’; *Il10* mutant reverse, 5’-CCACACGCGTCACCTTAATA-3’. 

### 2.2. Cecal Bacterial Lysate (CBL)

As an in vitro physiological bacterial stimulator, cecal bacterial lysate (CBL) was prepared from cecal contents from SPF C57BL/6J WT mice as described previously [35,36]. In brief, cecal contents were disrupted with 0.1-mm glass microbeads (BioSpec Products, Bartlesville, OK, USA) in MD solution containing 0.1 M magnesium chloride (Sigma-Aldrich, St. Louis, MO, USA) and 0.1 mg/mL DNase I (Worthington Biochemical, Lakewood, NJ, USA) by using a bead beater (Bio-Rad Laboratories, Hercules, CA, USA) and were centrifuged at 10,000 rpm for 15 min at 4 °C. The supernatants were filtrated through 0.45-µm filters (Genesee Scientific, San Diego, CA, USA). Sterility of the lysates was verified by YCFA [37] agar cultures at 37 °C for 5 days aerobically and anaerobically (Whitley MG500 workstation, N_2_:H_2_:CO_2_ = 80:10:10, Don Whitley Scientific, West Yorkshire, UK). Protein concentrations were measured by Bradford method with Bio-Rad protein assay kit (Bio-Rad Laboratories).

### 2.3. Cell Isolation from Spleen, MLN and cLP

Spleens and mesenteric lymph nodes (MLNs) were mechanically dissociated in complete medium RPMI1640 (Gibco/Life Technologies, Carlsbad, CA, USA) containing 5% heat-inactivated fetal bovine serum (FBS) (Millipore-Sigma, Burlington, MA, USA), 100 U/mL penicillin–streptomycin (Gibco/Life Technologies), 55 µM 2-mercaptoethanol (Sigma-Aldrich) and 1 mM sodium pyruvate (Gibco/Life Technologies). Red blood cells (RBCs) in spleen samples were lysed with red blood cell lysing buffer (Sigma-Aldrich). Cell preparations were filtrated through 70-μm nylon mesh (Fisher Scientific, Pittsburgh, PA, USA) to achieve single-cell suspensions. For separation of colonic lamina propria (cLP) cells, colons were opened longitudinally and washed with cold PBS (Mediatech, Manassas, VA, USA) to remove luminal content and mucus. After washing, colons were cut into 1-cm pieces and incubated in Hanks’ balanced salt solution (HBSS) (Mediatech) containing 4 mM EDTA (Mediatech), 10 mM dithiothreitol (Sigma-Aldrich), 2.5% FBS and 100 U/mL penicillin–streptomycin for 20 min at 37 °C with stirrer for 250 rpm to remove mucus and epithelial cells. After washing, the denuded tissues were digested with HBSS containing 0.5 mg/ml collagenase type IV (Sigma-Aldrich), 2.5% FBS and 100 U/mL penicillin–streptomycin for 30 min at 37 °C with stirrer for 350 rpm. After digesting tissues, the cLP cell suspensions were filtered through 100-μm nylon mesh (Fisher Scientific) and purified using a 40–70% discontinuous percoll gradient (20 min, room temperature, GE Healthcare, Piscataway, NJ, USA). Obtained cLP cells were washed with cold HBSS to remove percoll. Viability of isolated spleen, MLN, cLP cells was shown to be greater than 90% viable by trypan blue dye exclusion.

### 2.4. Cell Purification of APCs, BMDMs, B Cells and Naïve T Cells

Antigen-presenting cells (APCs) were separated from spleens from *Il10*^−/−^; *Rag2*^−/−^ double-deficient mice as described previously [36]. In brief, after RBC-lysis, spleen cells were overnight pulsed with 10 µg/mL CBL from C57BL/6J WT mice in complete medium at 37 °C with 5% CO_2_. After pulsing, APCs were collected and washed twice with PBS to remove soluble bacterial components and cytokines. Bone marrow-derived macrophages (BMDMs) were obtained as previously described [32]. B cells were purified magnetically by positive selection with anti-CD19 microbeads (Miltenyi Biotec, San Diego, CA, USA) after negative selection by a mixture of anti-CD90.2, anti-CD11c and anti-Ter119 microbeads (Miltenyi Biotec) according to the manufacturer’s instructions. Final B cell fractions were confirmed to be greater than 99.5% pure by flow cytometry, while cell viability was shown to be greater than 95% by trypan blue dye exclusion. Naïve CD4^+^ T cells (CD44 ^neg^CD62L^+^CD4^+^ T cells) were magnetically purified by naïve CD4^+^ T cells isolation kit (negative separation with anti-CD8a, CD11b, CD11c, CD19, CD25, B220, CD49b, CD105, MHC class II, Ter-119, TCR-γ/δ, and CD44 microbeads, Miltenyi Biotec). Purity of naïve CD4^+^ T cells was confirmed to be greater than 94.7% by flow cytometry using anti-CD3, TCR-β, CD4, CD8, CD44, and CD62L antibodies, while cell viability was shown to be greater than 95% by trypan blue dye exclusion.

### 2.5. Cell Culture with Bacterial Products and PI3K-Related Inhibitors

1 × 10^6^ unfractionated cells from spleen, MLN or cLP and colonic B cells were cultured in complete medium in 96-well round-bottom plates (Costar, Washington DC, MA, USA) for 24 h with or without the following bacterial products and PI3K-related inhibitors: 10 μg/mL CBL; 100 ng/mL Pam3CSK4 (Pam, InvivoGen, San Diego, CA, USA); 200 ng/mL lipopolysaccharide (LPS, InvivoGen); 500 nM CpG-DNA (CpG, InvivoGen); 2 μm PI3Kδ-selective inhibitor IC87114 (Selleck chemicals, Houston, TX, USA); 2 μm PI3K-global inhibitor LY294002 (Selleck chemicals) and 0.5 μm Wortmannin (Sigma-Aldrich); 0.5 μm Akt inhibitor API-2 (Bio-Techne, Minneapolis, MN, USA); dimethyl sulfoxide (DMSO, Fisher Scientific) as a vehicle. The doses of the above stimulants and inhibitors were chosen based on previous findings [6,11,32,36] and dose-related toxicity test cultures (cell viability of wild-type splenocytes after 1-day culture with a range of doses 0.1, 0.5, 1, 2, 5, 10, 20 µM of individual inhibitors). We chose the highest effective, but nontoxic doses of each inhibitor that resulted in a range of 75–82% cell viability, which were similar to 78% cell viability of the DMSO control group. Following cell cultures, supernatants were collected for measurements of cytokines by ELISA, while cells were analyzed by flow cytometry.

### 2.6. B Cell Co-Culture Assays with CD4^+^ T Cells and Macrophages

2 × 10^5^ naïve CD4^+^ T cells plus 2 × 10^5^ APCs were cultured with 5 × 10^5^ B cells in 96-well plates for 24 h. In another experiment, 1 × 10^5^ J774 A.1 macrophage cell line (TIB-67, passage 10–15, ATCC, Gaithersburg, MD, USA) or BMDMs were cultured with 5 × 10^5^ B cells, or LPS or CpG-stimulated B cell supernatants in 48-well plates for 72 h. These cells were selectively stimulated with either 10 μg/mL CBL, 100 ng/mL Pam, 200 ng/mL LPS or 500 nM CpG in complete medium. Following cell cultures, supernatants were collected for measurements of cytokines by ELISA.

### 2.7. T cell Proliferation Assay

Splenic naïve CD4^+^ T cells isolated from C57BL/6J WT mice were labeled with CFSE (BD Biosciences, San Jose, CA, USA) according to the manufacturer’s instructions. The 5 × 10^5^ CFSE-labeled naïve CD4^+^ T cells were co-cultured with or without 1 × 10^6^ splenic B cells isolated from WT or PI3Kδ^D910A^ mice with stimulation by 10 µg/mL anti-CD3e monoclonal antibody, 2 µg/mL anti-CD28 monoclonal antibody (Appendix A) and 10 µg/mL CBL for 72 h. PI3Kδ-selective inhibitor IC-87114 was selectively added. The proportion of CD4^+^CFSE^+^ cells was analyzed by flow cytometry.

### 2.8. Adoptive Transfer of Naïve CD4^+^ T Cells and CD19^+^ B Cells

Recipient *Il10*^−/−^; *Rag2*^−/−^ double-deficient mice were injected intraperitoneally with 5 × 10^5^ splenic naïve CD4^+^ T cells isolated from WT mice along with or without 1 × 10^6^ splenic B cells from WT, PI3Kδ^D910A^ or *Il10*^−/−^ donors. Six weeks after cell transfer, mice were sacrificed and colitis severity was evaluated by fecal lipocalin-2, histology scoring, and gene expression of proinflammatory cytokines in tissues.

### 2.9. Preparation for Quantification of Fecal Lipocalin-2

Fecal samples (10–20 mg) were incubated overnight at 4 °C in PBS containing 0.1% Tween 20 (Fisher Scientific) and vortexed shortly to obtain homogenous fecal suspensions. The fecal suspensions were centrifuged for 10 min at 12,000 rpm at 4 °C. Clear supernatants were collected and stored at −20 °C.

### 2.10. ELISA

Lipocalin-2 (Lcn) levels in fecal supernatant and cytokine levels in cell cultures were determined by ELISAs in duplicate, according to the manufacturer’s protocols (Lcn, IL-10, IL-17a, IFN-γ, TNF-α and IL-12p40 non-allele specific: R&D Systems, Minneapolis, MN, USA).

### 2.11. Quantitative PCR

The tissue RNA extractions were performed with RNeasy Plus Mini kit (Qiagen, Germantown, MD, USA) according to the manufacturer’s protocols. cDNA was created with the SensiFAST cDNA synthesis kit (Bioline, Memphis, TN, USA) by PCR (25 °C, 10 min; 42 °C, 15 min; 85 °C, 5 min). Quantitative PCR were performed with QuantStudio3 (Thermo Fisher Scientific, Pittsburgh, PA, USA) using SYBR No-ROX reagents (Bioline) with the following PCR setting: 95 °C, 2 min; 95 °C, 5 s; 40 cycles of (60 °C, 10 s; 72 °C, 20 s); melting curve analysis: 95 °C, 15 s; 60 °C, 15 s; 95 °C, 15 s. The data were created by comparative Ct method (2^−ΔΔCt^). The following PCR primers were used: *Il12b* (5’-CGCAAGAAAGAAAAGATGAAGGAG-3’) and (5’-TTGCATTGGACTTCGGTAGATG-3’); *Ifng* (5’-CTTCCTCATGGCTGTTTCTGG-3’) and (5’-ACGCTTATGTTGTTGCTGATGG-3’); *Il17a* (5’-CTCAGACTACCTCAACCGTTC-3’) and (5’-TGAGCTTCCCAGATCACAGAG-3’); *Tnfa* (5’-ACCCTCACACTCAGATCATCTTCTC-3’) and (5’-TGAGATCCATGCCGTTGG-3’); *Actb* (5’-AGCCATGTACGTAGCCATCCAG-3’) and (5’-TGGCGTGAGGGAGAGCATAG-3’). Each cDNA sample was analyzed in duplicate for quantitative assessment of RNA amplification. Melting curve analysis confirmed the presence of single products with expected melting temperatures.

### 2.12. Flow Cytometry

For flow cytometric analysis, single cells were incubated with anti-CD16/CD32 (BD Biosciences) as a Fc block for 10 min at 4 °C and then stained with fluorochrome-conjugated antibodies and proper isotype controls (Appendix A) for 20 min at 4 °C. Cells were washed and resuspended in PBS containing 1% bovine serum albumin (BSA) and then analyzed on a LSRII flow cytometer with FACSDiva software version 6.0 (BD Biosciences). Singlet live CD45^+^ cells were analyzed by FlowJo software version 10 (FlowJo, Ashland, OR, USA) with the following gating strategy: B cell (B220^+^CD19^+^), CD4^+^ T cell (TCRβ^+^CD3^+^CD4^+^CD8^neg^), CD8^+^ T cell (TCRβ^+^CD3^+^CD4^neg^CD8^+^), natural killer (NK) cell (TCRβ^neg^NK1.1^+^), macrophage (TCRβ^neg^CD11b^+^CD64^+^), Neutrophil (TCRβ^neg^MHCII^neg^Ly6G^+^) and dendritic cell (TCRβ^neg^CD64^neg^MHCII^+^CD11c^+^). When GFP was assessed, WT cells (GFP-negative) were stained with all antibodies used in the experiment except for the FITC/GFP channel, which is equivalent to a fluorescence-minus-one control [6].

### 2.13. Histological Colitis Score

Intestinal tissues were removed and fixed in 10% buffered formalin. Paraffin-embedded sections (5 µm) were prepared and stained with hematoxylin and eosin (H&E) by the Histology Core of the Center for Gastrointestinal Biology and Disease at UNC. The scoring of mucosal inflammation in cecum, proximal colon and distal colon was performed in a blinded fashion, with each region being graded from 0 to 4 as described previously [35]. The total histology scores represent the summation of all the scores (maximum score of 12).

### 2.14. Statistical Analysis

Statistical analysis was performed with Prism 8 software (GraphPad, San Diego, CA, USA). Significance between two groups was determined by unpaired Mann–Whitney test, while significance between more than 3 groups was determined by analysis of variance (ANOVA), followed by Dunn’s multiple comparisons test. For all statistical comparisons, *P* values less than 0.05 were considered significant.

## 3. Results

### 3.1. Inflammatory (IL-12p40) or Regulatory (IL-10) Pathways are Determined by PI3Kδ Signaling When Stimulated by Resident Bacteria

We first confirmed that PI3Kp110δ^D910A/D910A^ mutant (PI3Kδ^D910A^) mice in SPF conditions spontaneously develop colitis beginning at the age of 14 weeks or older based on serial fecal lipocalin-2 (f-Lcn) levels (Figure 1a) [38]. Histology assessment indicated that 8 week old PI3Kδ^D910A^ mice did not show any mucosal inflammation, while 18 week old PI3Kδ^D910A^ mice developed colitis characterized by mucosal crypt hyperplasia, lamina propria (LP) infiltration by neutrophils and lymphocytes and increased intraepithelial lymphocytes (Figure 1b,c) [12,31,32]. As inflammation modifies mucosal immune status [39,40,41], 8–10 week old mice were predominantly used for in vitro experiments. The size of MLNs and spleens are smaller and the length of colons is shorter in 8–10 week old PI3Kδ^D910A^ mice compared to WT mice (Appendix A). The cell numbers of PI3Kδ^D910A^ mice were decreased in the spleen, MLN and colonic LP (cLP) with striking reduction of frequencies in B cells (Appendix A), which might be due to impaired cell proliferation, differentiation or survival in PI3Kδ^D910A^ mice [31,42].

Given that PI3Kδ^D910A^ mice require the presence of resident microbiota to develop chronic colitis [12,32] similar to *Il10*^−/−^ mice [43], we next investigated immune responses against bacterial products in WT and PI3Kδ^D910A^ mice in vitro. Pam (TLR 1/2), LPS (TLR 4) and CpG-DNA (TLR 9) were selected based on the previous observations that these TLR ligands efficiently activate B cells [12,20,21,22,23,24,25,32]. We also utilized cecal bacterial lysates (CBL) as a physiological bacterial stimulant [35,36]. WT MLN cells produced abundant IL-10 when stimulated with any of the bacterial ligands tested, while PI3Kδ^D910A^ MLN cells secreted substantially decreased amounts of IL-10 (Figure 1d). Conversely, bacterial product-stimulated PI3Kδ^D910A^ MLN cells produced relatively high IL-12p40 levels, a subunit of pro-inflammatory cytokines IL-12 and IL-23 that promotes Th1/Th17 responses [24], although levels were not as pronounced as by *Il10*^−/−^ MLN cells (Figure 1e). These results indicate that a dysregulated immune response against bacterial product stimulation may contribute to mucosal inflammation in PI3Kδ^D910A^ mice.

We next performed a pharmacologic blockade of PI3K signaling on WT cells during bacterial product stimulation. IL-10 secretion by CBL-stimulated WT cLP, MLN and spleen cells was significantly inhibited by PI3K-global inhibitors (LY294002 and Wortmannin), PI3Kδ-selective inhibitor (IC87114) and an inhibitor of Akt, a molecule downstream of the PI3K pathway (Figure 1f and Appendix A) [44]. However, these inhibitors had no effect on IL-12p40 secretion. The IL-10/IL-12p40 ratio, which in part reflects the regulatory activity of immune cells, was significantly reduced by the PI3K/Akt inhibitors (Figure 1f and Appendix A). As cell viability confirmed by the trypan blue exclusion test was not different between the groups (data not shown), the limited IL-10 production by the PI3K/Akt inhibitors was neither due to the cytotoxicity nor induction of apoptosis by the inhibitors. The lack of suppression of IL-12p40 further confirmed the viability of cells after inhibitor treatment. Together, PI3K-Akt signaling appears to skew the bacterial-stimulated immune phenotype toward regulatory (IL-10) rather than inflammatory (IL-12p40) responses.

### 3.2. Colonic IL-10-Producing B Cells Are Decreased in PI3Kδ^D910A^ Mice

B cells are one of the key protective immune cells through IL-10 secretion in the pathogenesis of IBD [6,9,10,11,14,17,45,46,47] and the frequency of B cells is markedly decreased in PI3Kδ^D910A^ mice (Appendix A). Thus we investigated how PI3Kδ signaling influences the kinetics and functions of intestinal IL-10-producing regulatory B cells using IL-10-eGFP reporter mice (normal PI3Kδ (WT); *Il10*^eGFP^ and PI3Kδ^D910A^; *Il10*^eGFP^). Impressively, the number of cLP IL-10-producing B cells from 8–10 week old PI3Kδ^D910A^; *Il10*^eGFP^ mice, which have not yet developed colitis, were lower by 8-fold compared with those in WT; *Il10*^eGFP^ mice (Figure 2a). This reduction was large compared with the other regulatory cells, since FOXP3^+^CD4^+^ regulatory T cells showed relatively modest reduction in PI3Kδ^D910A^ cLP (2-fold) (Figure 2b). In contrast, 16–18 week old PI3Kδ^D910A^; *Il10*^eGFP^ mice with inflamed colons did not show significant reduction of cLP IL-10-producing B cell numbers (Figure 2c) or mean fluorescent intensity (data not shown), likely because inflammation induces IL-10 production by B cells [40,45].

In vitro, B cells were an important source of bacterial-stimulated IL-10 production (Figure 2d), and the frequencies of IL-10-producing B cells in PI3Kδ^D910A^ MLN cell cultures following bacterial stimulation were significantly lower compared to those in WT cell cultures (Figure 2d,e). Of note, IL-10-producing B cells produced more IL-10 in response to bacterial stimulation than did regulatory T cells both in vivo (Figure 2a,b) and in vitro (Figure 2f). To further analyze the association of the PI3Kδ pathway in intestinal IL-10-producing B cells, cLP B cells from WT mice were cultured with PI3K/AKT inhibitors in the presence or absence of bacterial stimulation. CBL promoted regulatory activity in B cells, indicated by an increased IL-10/IL-12p40 ratio, while blockade of PI3K-global or PI3Kδ-specific pathways significantly reversed this effect by selectively inhibiting secretion of IL-10 but not IL-12p40 (Figure 2g). These observations suggest that the PI3K signaling pathway is important for the IL-10-mediated regulatory function of bacteria-induced IL-10-producing B cells. In addition, we observed relatively lower fluorescence intensity of GFP (IL-10) in PI3Kδ^D910A^; *Il10*^eGFP^ B cells compared to WT; *Il10*^eGFP^ B cells (Appendix A), indicating functionally low activity of IL-10-producing PI3Kδ^D910A^ B cells. To identify the mechanisms underlying the lower IL-10 secreting ability in PI3Kδ^D910A^; *Il10*^eGFP^ B cells, we examined the surface markers in intestinal B cells. As we demonstrated recently, the phenotype of intestinal regulatory B cells in WT mice is IgM^low^IgD^low^CD23^low^CD24^high^ [6]. In contrast, gut PI3Kδ^D910A^; *Il10*^eGFP^ B cells show different expression patterns characterized by higher levels of IgM, IgD, and CD23, and lower expression of CD24 compared to WT B cells (Appendix A), indicating potentially different responses to stimuli. Together, these results suggest that a reduced number and functional properties of IL-10-producing B cells may contribute to the pathogenesis of intestinal inflammation in PI3Kδ^D910A^ mice.

### 3.3. Bacteria-Stimulated PI3Kδ^D910A^ B Cells Do Not Suppress Pro-Inflammatory Cytokine Secretion by Macrophages

Intestinal macrophages are a major source of cytokines in IBD [1,3,4]. To investigate the anti-inflammatory function of regulatory B cells on bacterial-activated macrophages, splenic B cells from either PI3Kδ^D910A^ or WT mice were co-cultured with the macrophage cell line J774, as a source of pro-inflammatory innate cytokines [1,3,4] (Figure 3a). PI3Kδ^D910A^ B cells produced less IL-10 than did WT B cells when stimulated by bacteria components (Figure 3b), with further induction of IL-10 by WT, but not PI3Kδ^D910A^, B cells in co-culture with J744 macrophages. Surprisingly, macrophages alone did not secrete IL-10 in the absence of B cells. LPS-, CpG- and CBL-induced IL-12p40 and TNF-α production by macrophages was significantly suppressed by co-cultured WT but not PI3Kδ^D910A^ B cells (Figure 3c,d). These results indicate that IL-10-producing B cells regulate macrophage responses to enteric bacterial ligands and that PI3Kδ signaling is required for B cell inhibition of macrophage activation by bacterial products.

### 3.4. Secreted IL-10 is the Dominant Way B Cells Regulate Pro-Inflammatory Cytokine Production in Macrophages in Response to Bacterial Products

We next sought to determine the mechanisms of WT B cells’ anti-inflammatory effects. Although humoral factors, especially secreted IL-10, are a key mediator of regulatory B cell activity direct cell-to-cell contact can also mediate B cells regulatory function [45,46]. Thus, we evaluated which factor is more predominantly involved in the bacteria-stimulated regulatory activity in B cells against inflammatory macrophages. To explore these possibilities, murine bone marrow-derived macrophages were cultured in the supernatants (excluding cells) from bacterial product-stimulated WT or PI3Kδ^D910A^ B cells (Figure 4a). As expected, IL-10 levels in PI3Kδ^D910A^ B cell supernatants were lower than those in WT B cell supernatants (Figure 4b). Pro-inflammatory IL-12p40 and TNF-α produced by the macrophages were significantly decreased with the supernatants from WT B cells, but only partially suppressed by the supernatants from PI3Kδ^D910A^ B cells (Figure 4c). Blockade of IL-10 signaling by anti-IL-10 receptor antibody negated the regulatory effect of the supernatant from WT B cells, indicating that the anti-inflammatory effect was primarily mediated by IL-10 (Figure 4d). These results suggest that incomplete suppression by the PI3Kδ^D910A^ B cell supernatant is primarily due to lower IL-10 concentrations. Moreover, as secreted IL-10 also affects B cells [45], we blocked IL-10 receptor on B cells to observe autocrine effect of IL-10. Interestingly, IL-10R blockade significantly increased IL-12p40 secretion in a dose-dependent manner in CpG-stimulated B cells (Appendix A). This finding indicates that low levels of IL-10 secretion from PI3Kδ^D910A^ B cells influence not only other surrounding cells but also the B cell responsiveness to bacterial stimuli. PI3Kδ signaling appears to modulate B cell phenotype both directly (by regulating TLR signaling) and indirectly (through an IL-10 autocrine pathway).

### 3.5. PI3Kδ-Signaling in B Cells Regulates Bacterial-Induced Pro-Inflammatory Cytokine Secretion by T Cells

As effector CD4^+^ T cells play a critical role in the pathogenesis of IBD [1,3,4], we explored the interactions between CD4^+^ T cells and B cells in intestinal inflammation. WT naïve CD4^+^ T cells plus *Il10*^−/−^ antigen presenting cells (APC) were co-cultured with WT, *Il10*^−/−^, or PI3Kδ^D910A^ B cells in the presence of CBL (Figure 5a). CBL activated naïve T cells to produce inflammatory IFN-γ and IL-17a in the presence of APC, while WT B cells [36], but not PI3Kδ^D910A^ B cells, produced abundant IL-10 and decreased T cell inflammatory cytokines (Figure 5b). Furthermore, WT B cells significantly suppressed naïve CD4^+^ T cell proliferation, while either PI3Kδ^D910A^ B cells or WT B cells plus PI3Kδ-inhibitors negated the suppressive effect of WT B cells on T cells (Figure 5c–e). These results indicate that B cells regulate CD4^+^ T cell proliferation and production of IFN-γ and IL-17 in response to bacteria and that PI3Kδ is required for these regulatory effects of B cells.

### 3.6. PI3Kδ-Signaling in B Cells is Required to Confer Protection Against T Cell-Mediated Colitis

Given the fact that the PI3Kδ pathway is involved in IL-10-mediated immune suppression by B cells in vitro, we investigated the involvement of B cell PI3Kδ in T cell-mediated colitis in vivo (Figure 6a). WT B cells decreased histologic inflammation in a CD4^+^ and B cell co-transfer colitis model with significantly less protection by either PI3Kδ^D910A^ or *Il10*^−/−^ B cells (Figure 6b,c). These results were confirmed by decreased levels of fecal Lcn (Figure 6d). Expression of colonic inflammatory cytokine genes, *Il12b*, *Ifng*, *Il17a* and *Tnfa* were significantly ameliorated by WT B cells but less-so by PI3Kδ^D910A^ and *Il10*^−/−^ B cells (Figure 6e). Together, these in vivo results indicate that PI3Kδ signaling plays a pivotal role in IL-10-producing B cell protection in T cell-mediated colitis.

## 4. Discussion

The present study demonstrates that a lack of PI3Kδ signaling decreases bacteria-stimulated IL-10 secretion by B cells and impairs B cell function to inhibit inflammatory responses in macrophages and T cells. Adoptively transferred PI3Kδ^D910A^ B cells were unable to attenuate T cell-mediated colitis. These findings indicate that PI3Kδ signaling is a key protective pathway for mucosal homeostasis and prevention of intestinal inflammation through bacterial-activated IL-10-producing B cells.

Dysfunction of the PI3Kδ pathway in mice leads to spontaneous colitis in the presence of resident bacteria and exacerbates mucosal inflammation in murine experimental colitis models [12,30,31,32]. In humans, the PI3Kδ gene (*PIK3CD*) maps to the IBD7 susceptibility locus on chromosome 1p36 [28,29], while the PI3Kδ-inhibitor Idelalisib, a therapeutic agent for certain tumors, can cause colitis as a side effect [48]. Based on our present results showing that B cell PI3Kδ plays an anti-inflammatory role in the activation of macrophages and T cells and can attenuate T cell-mediated colitis, these clinical and basic observations might be in part due to PI3Kδ-mediated dysfunction of IL-10-producing regulatory B cells. Despite previous evidence showing dysfunction of multiple types of innate immune cells in PI3Kδ^D910A^ mice [12,32], the present study showed that defective PI3Kδ signaling decreased the frequency and function of regulatory B cells in response to bacterial products to a greater extent than that of regulatory T cells or macrophages. Blockade of PI3Kδ signaling caused B cells to lose their ability to control bacterial-activated inflammatory cytokine production by macrophages or T cells. Finally, intact B cell-specific PI3Kδ signaling was sufficient to significantly ameliorate colitis in a T cell-transfer colitis model. These findings indicate that PI3Kδ signaling in B cells is a key determinant of the homeostatic phenotype of the mucosal immune system, and suggests that restoring or stimulating the PI3Kδ pathway in B cells might be a therapeutic target in IBD.

Importantly, others have demonstrated contradictory roles for PI3Kδ in inflammatory diseases outside the intestine. Blockade of PI3Kδ attenuates inflammation in arthritis in K/BxN serum-transferred mice [49], IgE- or ovalbumin-related allergic models [50,51,52,53], a multiple sclerosis model [54], imiquimod-induced dermatitis [55], and experimental systemic lupus erythematosus in BXSB mice [56]. Inhibition of PI3Kδ signaling may predominantly suppress aggressive inflammatory immune cells over regulatory immune cells in non-IBD models, where resident intestinal bacteria are not the dominant activators of inflammation. Supporting this concept, pathogenic function of effector cells, such as neutrophils [49,51], CD8^+^ T cells [50,51], mast cells [50,51,53], Th2 cells [50,51,52], CD44^high^CD62L^low^CD4^+^ T cells [54], Th17 cells [54,55,56], and autoreactive immunoglobulin-secreting B cells [56], were decreased by PI3Kδ signaling blockade in experimental inflammation outside the intestine. In contrast, aberrant activation of innate immune cells (macrophages, dendritic and antigen-presenting cells) and Th1/Th17 cells, in concert with decreased regulatory cells (regulatory T cells and B cells and intestinal dysbiosis), strongly contribute to the onset and progression of IBD and experimental intestinal inflammation [2,3,5,6,9,10,11,14,17,45,46,47]. Interestingly, human regulatory T cells, compared to CD4^+^ or CD8^+^ T cells, were shown to be exquisitely sensitive to the effects of specific PI3Kδ inhibition in terms of proliferation and suppressive function [57]. Another newly described immune deficiency/inflammatory condition termed activated PI3Kδ syndrome (APDS) has emerged and contributed additional insight into the immunoregulatory role of PI3Kδ in human diseases [58]. Patients with APDS have a gain of function mutation in the gene encoding PI3Kδ (*PIK3CD*) and present with recurrent bacterial and herpesvirus sinopulmonary infections, lymphadenopathy, and autoimmune/inflammatory manifestations. It is tempting to speculate that patients with APDS demonstrate opposing immune phenotypes to that of patients with IBD, due to suppression of normal bactericidal/antiviral immune responses at mucosal surfaces. Ultimately, whether through inadequate pathogen clearance (in APDS) or prolonged inappropriate inflammatory response (in IBD), these two conditions both lead to aberrant chronic inflammation and its sequelae. These data suggest that defective PI3Kδ signaling in response to resident bacteria might influence regulatory immune cells to a greater degree than effector immune cells in the intestine and other mucosal surfaces regularly in contact with resident microbes.

We also emphasize the importance of colonic lamina propria (cLP) IL-10-producing cells in intestinal homeostasis. cLP cells demonstrated markedly higher production of IL-10 and regulatory activity (IL-10/IL-12p40 ratio) compared with MLN and spleen cells in response to bacterial stimulation. The colon is an unique environment where constant exposure to resident bacteria and their products requires dedicated regulation via IL-10 to maintain homeostasis. Likewise, the colon may be more sensitive to defective IL-10 signaling compared with other organs. Our data showed that dysregulated immunity in PI3Kδ^D910A^ mice predominantly develops in the colon through diminished IL-10 production. PI3Kδ^D910A^ mice develop colitis characterized by microbiota-dependent Th1/Th17 cell expansion [12,31,32] despite low IFN-γ production by antigen-stimulated PI3Kδ^D910A^ T cells [59]. This discrepancy remains unclear, but our co-culture assays revealed that defective PI3Kδ signaling in B cells promoted low IL-10 and high IL-12p40 production, impaired suppression of IL-12p40 secretion by bacteria-stimulated macrophages, and promoted naïve T cell proliferation. Since IL-12p40 promotes differentiation of naïve T cells into Th1/Th17 cells [24,60], prolonged APC stimulation conditions (high IL-12p40 and low IL-10) by PI3Kδ^D910A^ B cells may promote expansion of Th1/Th17 cells. Moreover, IL-12p40 and the Th1 cytokine IFN-γ normally induce anti-inflammatory cytokines from T cells and B cells and regulatory cells as a negative feedback [40,45,61,62,63], but IL-10-producing regulatory B cells are not appropriately induced to make IL-10 in PI3Kδ^D910A^ mice (age older than 14 weeks). These findings indicate that homeostatic IL-10-production by B cells and IL-10^+^ regulatory B cell numbers are not adequately induced under excessive inflammatory conditions in the absence of PI3Kδ signaling, which likely worsens mucosal inflammation. Together, a deregulated immune phenotype (low IL-10 and high IL-12p40) of PI3Kδ^D910A^ B cells appears to contribute to the pathogenesis of T cell-mediated colitis.

One of the limitations of the present study is that we did not assess IL-10-independent regulatory factors of PI3Kδ B cells, which include TGF-β, IL-27, IL-35, IgA and induction of regulatory T cells [6,9,10,11,45,46,47,64]. However, our in vitro assays with an anti-IL-10 receptor antibody demonstrate the importance of IL-10 produced by B cells in regulating inflammatory cytokines from macrophages, T cells, and B cells themselves. Given that deficiency of IL-10, PI3Kδ, or luminal IgA affects gut microbiota colonization [65,66,67], profiling enteric microbiota in PI3Kδ^D910A^ mice may unveil a mutual relationship between PI3Kδ function in B cells and microbial dysbiosis in IBD. An additional limitation is that we were unable to determine which bacterial species, groups or components are responsible for stimulating IL-10 through PI3Kδ in the murine B cells with our cecal lysates. This can be addressed in future studies by exploring the relative efficacy of lysates of relevant individual resident intestinal bacterial species or their components. Finally, future studies should test the ability of human fecal samples to activate PI3Kδ-dependent protective functions in human-derived colonic immune cells.

In conclusion, our study provides evidence that genetic dysfunction or pharmacological blockade of PI3Kδ in B cells lead to globally dysfunctional homeostatic pathways in the intestine and consequent development of intestinal inflammation. As PI3K signaling affects several key immune pathways, investigating cell-specific, site-specific and disease-specific PI3K-related signaling is important for better understanding of the pathogenesis of IBD and regulation of mucosal homeostasis. This knowledge may contribute to the development of novel and safe pharmacological therapies in IBD.

## Figures and Tables

**Figure 1 cells-08-01121-f001:**
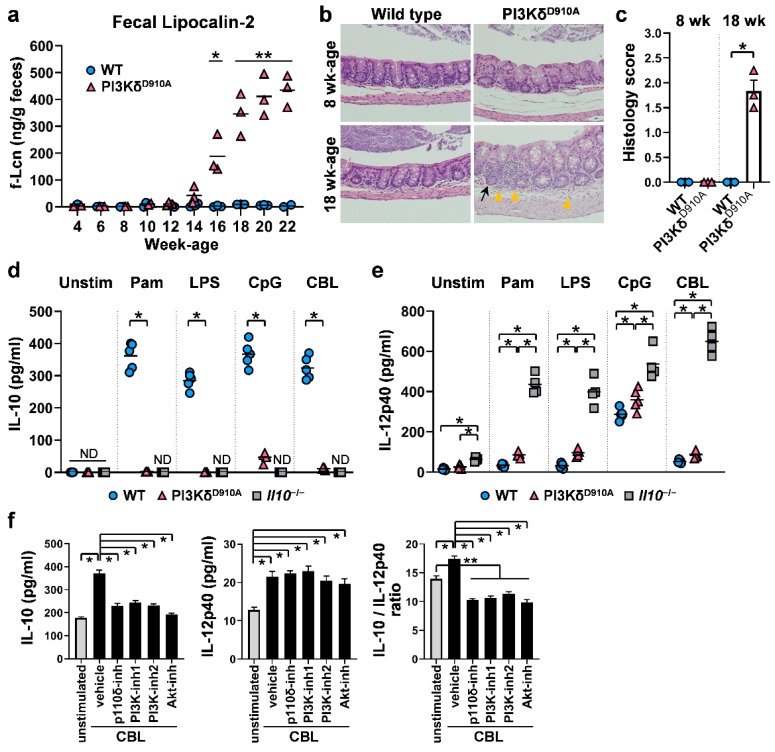
Inflammatory or regulatory phenotypes are determined by PI3Kδ signaling. (**a**) The time course of concentrations of fecal lipocalin-2 (f-Lcn) from wild-type (WT) and PI3Kδ^D910A^ mice. (**b**) Representative images of H&E-stained cecal samples from C57BL/6 wild-type mice (WT) or PI3Kp110δ^D910A/D910A^ mice (PI3Kδ^D910A^) at 8 and 18 weeks of age. The scale bars represent 100 µm. Black arrow indicates crypt hyperplasia. Yellow arrowhead indicates infiltration of neutrophils and lymphocytes. (**c**) Histology scores of WT or PI3Kδ^D910A^ mice at 8 and 18 weeks of age are shown. (**d,e**) 1 × 10^6^ unfractionated mesenteric lymph node (MLN) cells from WT, PI3Kδ^D910A^ or IL-10-deficient (*Il10*^−/−^) mice were cultured with medium alone (unstim), Pam3CSK4 (Pam, 100 ng/mL), lipopolysaccharide (LPS, 200 ng/mL), CpG-DNA (CpG, 500 nM) or cecal bacterial lysate (CBL, 10 µg/mL) in complete medium for 24 h at 37 °C with 5% CO_2_. Supernatant levels of (**d**) IL-10 and (**e**) IL-12p40 were quantified by ELISA. (**f**) 1 × 10^6^ unfractionated cLP from WT mice were cultured with or without CBL (10 µg/ml) in the absence or presence of specific PI3K/AKT-related inhibitors or vehicle control in complete medium for 24 h. IL-10 and IL-12p40 levels in cell culture supernatants were measured by ELISA and IL-10/IL-12p40 ratios were calculated. PI3K/Akt-related inhibitors: PI3Kp110δ-selective inhibitor (p110δ-inh, IC87114, 2 µM), PI3K-global inhibitors (PI3K-inh1: LY294002, 2 µM and PI3K-inh2: wortmannin, 0.5 µM); Akt inhibitor (Akt-inh, API-2, 0.5 µM). Vehicle control: dimethyl sulfoxide. N = 4-5/group. Each symbol represents the result from individual mice and bars indicate the mean in **a**,**d**–**e**. Mean ± SEM are shown in **c**,**f**. Mann–Whitney unpaired two-tailed test was used for **a**,**c**. Dunn’s multiple comparisons test following one-way ANOVA was used for **d**–**f**. **p* < 0.05, ***p* < 0.01. ND indicates not detected.

**Figure 2 cells-08-01121-f002:**
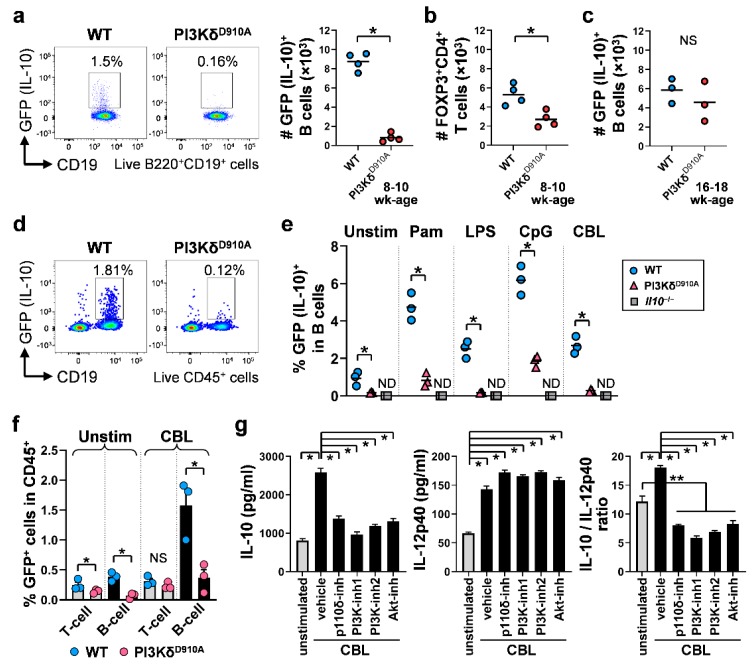
IL-10-producing B cells are significantly decreased in the colon of PI3Kδ^D910A^ mice. (**a**) Representative dot plots and frequencies of GFP (IL-10)^+^ cLP B cells (Live/Dead^neg^CD45^+^B220^+^CD19^+^) from 8–10-week-old normal PI3Kδ (WT); *Il10*^eGFP^ mice or PI3Kδ^D910A^; *Il10*^eGFP^ mice. Cell numbers are presented per 1 × 10^6^ live cLP cells (Live/Dead^neg^CD45^+^). (**b**) The abundance of cLP FOXP3^+^CD4^+^ T cells (Live/Dead^neg^CD45^+^TCRβ^+^CD3^+^CD4^+^CD8^neg^FOXP3^+^) is demonstrated from 8–12-week-old WT; *Il10*^eGFP^, PI3Kδ^D910A^ or *Il10*^−/−^ mice. (**c**) GFP^+^ B cells from 16–18-week-old WT; *Il10*^eGFP^ or PI3Kδ^D910A^; *Il10*^eGFP^ mice are shown. (**d**–**f**) 1 × 10^6^ unfractionated MLN cells from 8–10-week-old WT; *Il10*^eGFP^, PI3Kδ^D910A^ or *Il10*^−/−^ mice were cultured with medium alone (unstim), Pam3CSK4 (Pam, 100 ng/mL), lipopolysaccharide (LPS, 200 ng/mL), CpG-DNA (CpG, 500 nM) or CBL (10 µg/mL) in complete medium for 24 h at 37 °C with 5% CO_2_. Representative dot plots of MLN cells stimulated with CBL are shown in d. The percentages of GFP (IL-10)^+^ in the total B cell population are shown in e. The percentages of GFP^+^ T cells (CD45^+^TCRβ^+^CD3^+^) and B cells (CD45^+^B220^+^CD19^+^) in MLN cells (CD45^+^) stimulated with CBL are shown in f. (**g**) 5 × 10^6^ cLP B cells magnetically isolated from WT mice were cultured with or without CBL (10 µg/mL) in the absence or presence of PI3K/Akt inhibitors or vehicle control as described in Figure 1 legend in complete medium for 24 h. IL-10 and IL-12p40 levels in supernatants were measured by ELISA and IL-10/IL-12p40 ratios were calculated. Each symbol represents the result from individual mice, and bars indicate the mean in a–c, and e. Mean ± SEM are shown in f,g. Mann–Whitney unpaired two-tailed test was used for a–c, e, one-tailed for f. Dunn’s multiple comparisons test following one-way ANOVA was used for g. **p* < 0.05, ***p* < 0.01. ND indicates not detected. NS indicates not significant.

**Figure 3 cells-08-01121-f003:**
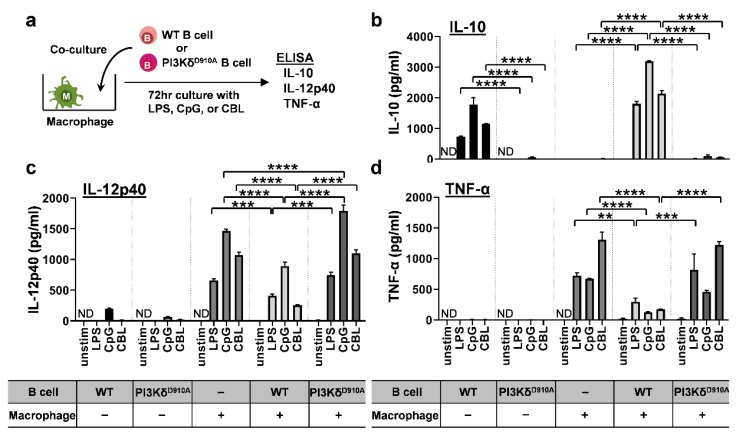
Bacteria-stimulated WT but not PI3Kδ^D910A^ B cells suppress IL-12p40 and TNF-α secretion by macrophages. (**a**) 1 × 10^5^ J774 macrophages were cultured with 5 × 10^5^ splenic B cells from C57BL/6 mice (WT) or PI3Kδ^D910A^ mice with various bacterial stimuli, LPS (200 ng/mL), CpG (500 nM), CBL (10 μg/mL) or medium alone (unstim) for 72 h in complete medium at 37 °C with 5% CO_2_. Supernatants were assessed for (**b**) IL-10, (**c**) IL-12p40, and (**d**) TNF-α by ELISA in duplicate. Mean ± SEM. N = 5–6/group. Dunn’s multiple comparisons test following 2-way ANOVA was used. ***p* < 0.01, ****p* < 0.001, *****p* < 0.0001. ND indicates not detected.

**Figure 4 cells-08-01121-f004:**
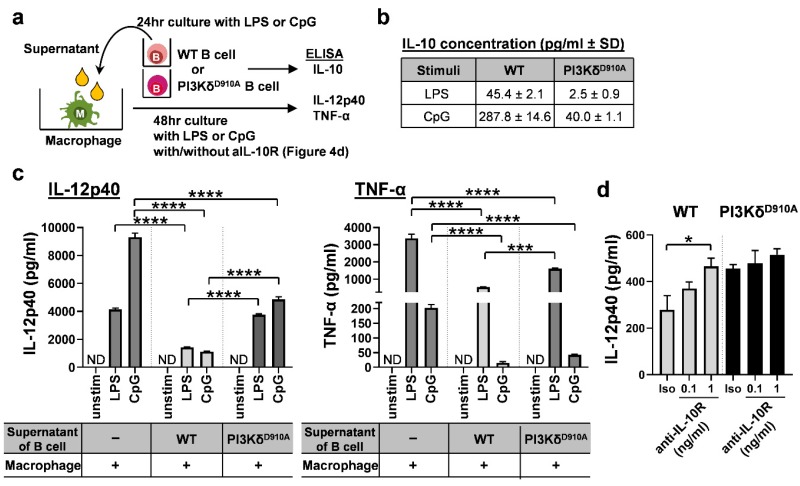
Secreted IL-10 from PI3Kδ^D910A^ B cells is not sufficient to regulate inflammatory cytokine production by macrophages in response to bacterial products. (**a**) 1 × 10^6^ splenic B cells from WT or PI3Kδ^D910A^ mice were cultured with LPS (200 ng/mL) or CpG (500 nM) for 24 h, then supernatants were harvested by spinning the culture cells down. (**b**) Concentrations of IL-10 in the supernatants were assessed by ELISA. The supernatants were added to 1 × 10^5^ bone marrow derived macrophage (BMDM) cultures with the same concentration of LPS or CpG for another 48 h. BMDM were derived from WT C57BL/6 mice. (**c**) The final supernatants were assessed for IL-12p40 and TNF-α by ELISA. (**d**) BMDMs were cultured in LPS (200 ng/mL)-containing supernatants from either WT B cells or PI3Kδ^D910A^ B cells with anti-IL-10R (0.1 or 1 ng/mL) or isotype IgG control (1 ng/mL) for 24 h. IL-12p40 level was assessed by ELISA. Mean ± SEM. Dunn’s multiple comparisons test following ANOVA was used for **c**, **d**. **p* < 0.05, ****p* < 0.001, *****p* < 0.0001. ND indicates not detected.

**Figure 5 cells-08-01121-f005:**
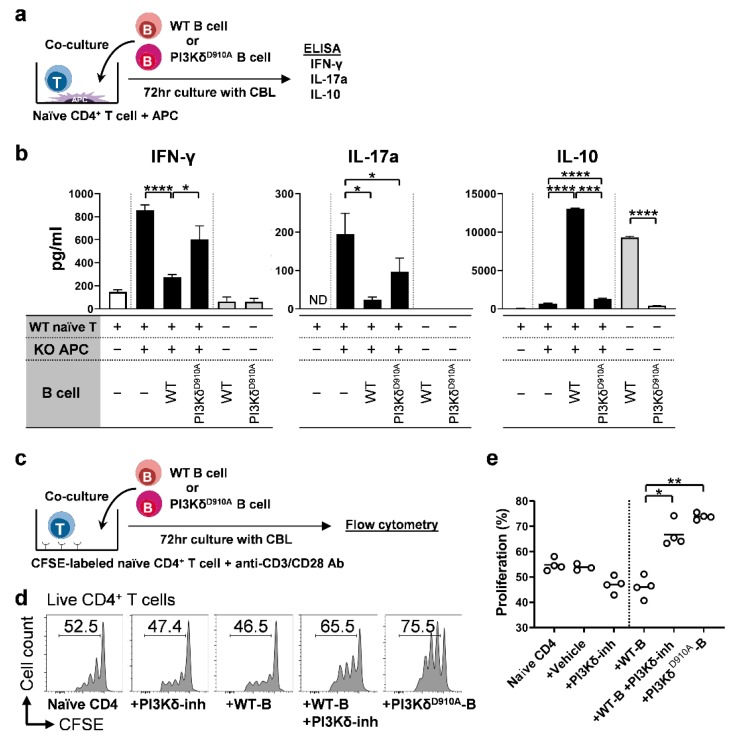
PI3Kδ-signaling in B cells regulates bacteria-stimulated inflammatory cytokine production by and proliferation of T cells. (**a**,**b**) 5 × 10^5^ WT splenic naïve CD4^+^ T cells plus 2 × 10^5^
*Il10*^−/−^; *Rag2*^−/−^ antigen-presenting cells (APC) were co-cultured with or without 1 × 10^6^ splenic B cells in the presence of 10 µg/mL CBL for 72 h. Supernatant levels of IL-10, IFN-γ and IL-17a were measured by ELISA in duplicates. N = 4–6/group. (**c**–**e**) The 5 × 10^5^ CFSE-labeled WT naïve CD4^+^ T cells were cultured with 1 × 10^6^ B cells from WT or PI3Kδ^D910A^ mice for 72 h with CBL (10 μg/mL). After culturing, single cells were stained and proliferation of naïve CD4^+^ T cells were assessed by flow cytometry. Representative histograms of live CD4^+^ T cells (Live/Dead^neg^CD45^+^TCRβ^+^CD3^+^CD4^+^CD8^neg^) are shown in **d**. Percentage of proliferating cells was demonstrated in **e**. Bars indicate the mean. Dunn’s multiple comparisons test following one-way ANOVA was used for **b**,**e**. **p* < 0.05, ***p* < 0.01, ****p* < 0.001, *****p* < 0.0001. ND indicates not detected.

**Figure 6 cells-08-01121-f006:**
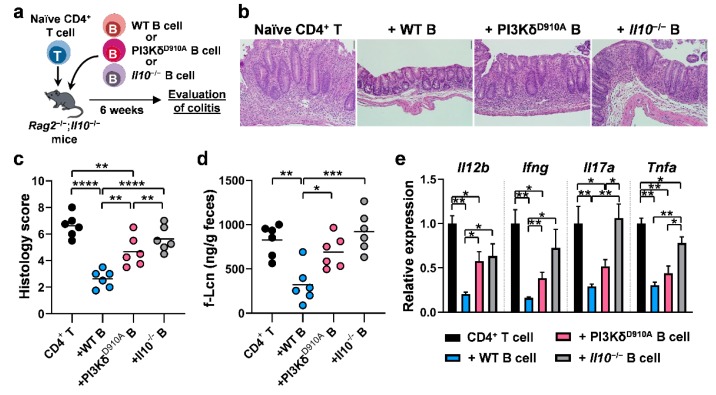
PI3Kδ-signaling in B cells is required to confer protection against T cell-mediated colitis. (**a**) 5 × 10^5^ splenic naïve CD4^+^ T cells were co-transferred intraperitoneally with or without 1 × 10^6^ splenic B cells from WT, PI3Kδ^D910A^, or *Il10*^−/−^ mice into *Rag2*^−/−^; *Il10*^−/−^ mice. The severity of colitis was evaluated 6 weeks after cell transfer. N = 6/group. (**b**) Representative images of H&E-stained cecum are shown. Scale bar represents 100 µm. (**c**) Histology scores are shown. Bar indicates the mean. (**d**) Lipocalin-2 in fecal supernatant (f-Lcn) was assessed by ELISA. Bar indicates the mean. (**e**) Expression of inflammatory cytokines in the distal colon was assessed by qPCR. Relative quantification was normalized to WT expression levels. Dunn’s multiple comparisons test following one-way ANOVA was used. **p* < 0.05, ***p* < 0.01, ****p* < 0.001, *****p* < 0.0001.

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
