# Peer review of "Phosphoinositide 3-Kinase P110δ-Signaling Is Critical for Microbiota-Activated IL-10 Production by B Cells that Regulate Intestinal Inflammation"

_cells, 2019, doi:10.3390/cells8101121_

Round 1
Reviewer 1 Report
This is an interesting manuscript that documents an important role for B cells in the intestinal inflammation that develops in mice with inactive PI3Kdelta, through a reduced production of the immunosuppressive cytokine IL-10. This is of direct relevance to human biology, given that patients treated with PI3Kdelta inhibitors frequently develop colitis.
The data in this MS are experimentally sound, and the manuscript is well-written and referenced.
I only have one major comment:
page 1, lines 23-24: it is an overstatement that polymorphisms in the PI3Kdelta gene are associated with IBD susceptibility. While the PIK3CD gene indeed maps to an IBD sensitivity locus, I do not recall formal evidence that PIK3CD polymorphism are indeed linked with IBD sensitivity. I think this is not more than a pure speculation at this point in time - this statement is therefore misleading and should be toned down. This also applies to page 13 (lines 417-418).
Reviewer 2 Report
This is a very interesting paper which aimed to study the involvement of phosphoinositide 3 kinase p110δ-signaling for regulation of microbiota-activated IL-10 production by B cells that plays an important role in modulation of intestinal inflammation. The studies were well designed which comprise different comprehensive in vitro and in vivo models to address different questions. This paper have no doubt provided additional knowledge to the field of immunology, in particular with potential application of the knowledge to the development of novel and safe pharmacological therapies to the inflammatory diseases like IBD. There are just few critical items that the authors need to address before publication can be considered.
It is advisable to state a clear hypothesis of the study in the introduction and then elucidate how the research objectives were designed to accomplish the hypothesis. Different inhibitors (e.g. CBL, Pam etc.) have been used by the authors. There is no information about how the doses of these inhibitors were chosen. Why J774 A.1, which is a cancerous cell line was chosen as one of the models? The paper is in general very descriptive in terms of the methods used. However, for qPCR, why only b actin was solely chosen? geNorm or NormFinder are commonly used to check if housekeeping gene chosen is stable among all treatments. Also the authors should provide some more information about the purity assessment of RNA for qPCR work. (e.g. RNA integrity, melting curve analysis for qPCR etc.) For statistical analysis, are the data normally distributed? If not, it has to be done and appropriate statistics has to be applied. For 2.14, the authors stated unpaired t test (parametric) was used but in Fig. 1, 2, Mann Whitney test (non-parametric) was mentioned. Also under what circumstances Dunn’s test or Tukey’s test was used? It is indeed worthy to examine the profile of enteric microbiota in PI3KδD910A mice by pyrosequencing in the future as suggested by the authors. However, since cecal bacterial lysate was used, there is also a limitation that one cannot know which specific type of bacterial plays the role in modulating inflammation. One may also suggests the use human fecal samples and immune cells isolated from human donors to confirm the findings obtained in this study?
